# The Application of Pureed Butter Beans and a Combination of Inulin and Rebaudioside A for the Replacement of Fat and Sucrose in Sponge Cake: Sensory and Physicochemical Analysis

**DOI:** 10.3390/foods10020254

**Published:** 2021-01-26

**Authors:** Aislinn M. Richardson, Andrey A. Tyuftin, Kieran N. Kilcawley, Eimear Gallagher, Maurice G. O’Sullivan, Joseph P. Kerry

**Affiliations:** 1Food Packaging Group, School of Food and Nutritional Sciences, University College Cork, College Road, Cork, Ireland; 108003904@umail.ucc.ie (A.M.R.); a.tiuftin@ucc.ie (A.A.T.); 2Sensory Group, School of Food and Nutritional Sciences, University College Cork, College Road, Cork, Ireland; 3Food Quality and Sensory Science Department, Teagasc Food Research Centre, Moorepark, Fermoy, Cork, Ireland; kieran.kilcawley@teagasc.ie; 4Ashtown Food Research Centre, Teagasc, Ashtown, D15, Dublin, Ireland; eimear.gallagher@teagasc.ie

**Keywords:** inulin, sugar reduction, sucrose replacement, fat reduction, Rebaudioside A, sponge cake, butter beans, sensory, physicochemical properties, bakery

## Abstract

Determining minimum levels of fat and sucrose needed for the sensory acceptance of sponge cake while increasing the nutritional quality was the main objective of this study. Sponge cakes with 0, 25, 50 and 75% sucrose replacement (SR) using a combination of inulin and Rebaudioside A (Reb A) were prepared. Sensory acceptance testing (SAT) was carried out on samples. Following experimental results, four more samples were prepared where fat was replaced sequentially (0, 25, 50 and 75%) in sucrose-replaced sponge cakes using pureed butter beans (Pbb) as a replacer. Fat-replaced samples were investigated using sensory (hedonic and intensity) and physicochemical analysis. Texture liking and overall acceptability (OA) were the only hedonic sensory parameters significantly affected after a 50% SR in sponge cake (*p* < 0.05). A 25% SR had no significant impact on any hedonic sensory properties and samples were just as accepted as the control sucrose sample. A 30% SR was chosen for further experiments. After a 50% fat replacement (FR), no significant differences were found between 30% sucrose-replaced sponge cake samples in relation to all sensory (hedonic and intensity) parameters investigated. Flavour and aroma intensity attributes such as buttery and sweet and, subsequently, liking and OA of samples were negatively affected after a 75% FR (*p* < 0.05). Instrumental texture properties (hardness and chewiness (N)) did not discriminate between samples with increasing levels of FR using Pbb. Moisture content increased significantly with FR (*p* < 0.05). A simultaneous reduction in fat (42%) and sucrose was achieved (28%) in sponge cake samples without negatively affecting OA. Optimised samples contained significantly more dietary fibre (*p* < 0.05).

## 1. Introduction

Sponge cake, which starts as a fluid batter, is classified by its solid porous structure after baking. It is prepared using four main ingredients, namely, eggs, sugar, wheat flour and fat, which are responsible for multiple functions in the batter and during baking. The presence of fat and sugar in food products yields positive hedonic responses which may override metabolic responses such as satiety [1].

According to WHO (World Health Organization) guidelines (2011), adults and children should consume only 10% of calories taken from sugars in their daily diet, which is about 10–14 teaspoons of sugar. In Ireland, this number achieved 14.6%, which is 31.5% higher than the recommended dosage [2]. This contributes to dietary imbalances associated with overweight and obesity and consequently the prevalence of non-communicable diseases [3]. As cakes contribute greatly to the dietary intake of sugar and fat in developed countries [4,5], it is necessary to find ways to reduce/replace sugar and fat in these products.

Before we consider sucrose and fat replacement/reduction strategies, we must first understand the functions performed by these ingredients in cakes. Sucrose is the most common sugar used in baking [6]. Sucrose is responsible for the tenderisation of cakes by competing with gluten for available water during batter mixing, thus impeding on the formation of a strong gluten network [7]. During baking, sucrose increases the temperature of starch gelatinisation and protein denaturation by binding water and therefore limiting water availability to starch granules [8]. In doing so, sucrose increases the volume and bulk of the final product. The most commonly known function of sucrose is that it provides sweetness to cakes and also acts a flavour enhancer [6]. In relation to the function of fats, fat entraps air in the batter during the creaming step which provides a structure for leavening gases [7]. Thus, fat also affects the volume and bulk of the final product. Fat tenderises and adds lubricity to the texture of cakes by coating the protein and starch particles, thereby interfering with the protein matrix [9]. Fat also emulsifies liquid in cake production which adds further moisture and softness to the product [10]. Fat sources such as butter contribute to flavour and can act as a flavour enhancer [10].

Thus, as sugar and fat perform many key functions in cakes, it is difficult to reduce and replace these ingredients without affecting the physical quality and important sensory attributes associated with liking and overall acceptability.

Several authors have studied the impact of sucrose replacement (SR) using a variety of different replacers such as polyols, fibres and artificial intense sweeteners on the overall quality of cakes [11]. Fibres such as inulin are of particular interest for use as sugar replacers as they possess prebiotic properties [12]. Inulin (Figure 1) is a plant-derived storage polysaccharide which is predominantly produced by chicory roots on a commercial basis.

Roberfroid [13] reported that fructans such as inulin may have positive effects regarding colon cancer prevention. Furthermore, the incorporation of dietary fibre into foods is now considered as a principal prevention strategy against the risk of non-communicable disease [14]. Fibres such as inulin provide the bulk properties (texture and volume) by binding water [15] and therefore competing with gluten proteins for available water and delaying starch gelatinisation. Inulin contains 25–35% of the energy of digestible carbohydrates and its sweetness level is 10% that of sucrose [12]. Cakes containing 30% sugar replacement (SR) with inulin were shown to be similar to control sugar samples with regard to bubble size distribution and physical and sensory properties in a study conducted by [16]. As mentioned, inulin has a low sweetness value relative to that of sucrose, and therefore it is necessary to combine inulin with an intense sweetener. In a study conducted by Struck et al. (2016) [11,17], a 30% SR in muffins with inulin and Rebaudioside A (Reb A) produced cakes with a similar descriptive sensory profile to reference muffins. Reb A is a steviol glycoside (Figure 2) whose sweetness is 300 times greater in comparison with a sucrose [18,19]. Reb A represents white odorless crystals. Steviol glycosides such as Reb A were approved by the European Union for use in foods and are considered natural intense sweeteners [20]. These sweeteners are a special class of intense sweeteners as they are considered natural unlike most other intense sweeteners [20] because they are produced from Stevia leaves [21].

Therefore, the present study investigated the impact of a sequential SR (0, 25, 50 and 75) using a combination of inulin and Reb A on the sensory acceptance of sponge cake to determine the minimum levels of sucrose needed in the formulation to maintain sensory acceptance.

In relation to the reduction in/replacement of fat in cake products, three main classes of fat substitutes exist, namely, carbohydrate-based, protein-based and fat-based [22]. Carbohydrates are used as fat substitutes as they are similar to triglycerides in relation to physical and chemical structure [23] and they bind water and therefore contribute to texture and mouthfeel. In a study conducted using three commercial carbohydrate-based fat replacers derived from pectin, gums and oat bran, moisture content increased significantly in biscuits with a fat substitution [23]. Legumes such as beans are also used as fat replacers as beans are a source of complex carbohydrates roughly containing up to 60% total carbohydrates. Legumes are low glycaemic and high in protein and fibre and their consumption has been associated with a decreased risk of coronary heart disease [24]. In a study conducted on brownies, pureed black beans successfully replaced fat in relation to sensory acceptance at a level of 30% replacement [25]. Furthermore, this study reported that a 90% substitution of fat with black beans still produced an acceptable product. In a study conducted on oatmeal cookies, white beans were successful in replacing fat up to a level of 25% without significantly affecting sensory properties [26]. Cannellini beans have been used to successfully replace fat in brownies by up to 50% without affecting OA, texture and flavour properties [27].

For the above reasons, the present study utilised pureed butter beans (Pbb) for the sequential replacement of fat (0, 25, 50 and 75%) in reduced sugar sponge cake. Determining minimum levels of both sugar and fat needed to maintain sensory acceptance is imperative and could be a significant development in reducing the dietary intake of both fat and sugar.

Thus, the main aim of this work was, firstly, to determine the minimum levels of sucrose needed in sponge cake to maintain sensory acceptance similar to that of the control sucrose formulation. This was determined through sensory acceptance testing (SAT). Secondly, and following experimental results, the aim was to determine the minimum levels of fat needed to maintain sensory properties associated with liking and OA of reduced sugar sponge cake. This was determined by SAT and optimised descriptive profiling (ODP). Other characteristics such as texture, colour and compositional properties were studied for cakes during sequential FR of sucrose-replaced sponge cake.

## 2. Materials and Methods

Food ingredients used in this trial included free-range eggs (Upton brand, Ireland); caster sugar (99.9% sucrose, 0.3% moisture and 0.01% sodium, Tate & Lyle brand, UK); cream plain flour (82.7% carbohydrate, 2% of which were sugars, 11.7% protein, 3.4% fibre, 1.4% fat and 0.81% salt, Odlums brand, Ireland); Irish creamery butter (81% total fat, 65.4% of which were saturated and 15.1% moisture, Dunnes stores brand, Ireland); inulin (89% fibre and 8% sugar, Bioglan brand, Australia); Reb A, (Bulk Powders brand, Ireland); butter beans (16.8% carbohydrate, 1.4% of which were sugars, 0.6% fats 0.1% of which were saturated, 5.2% fibre, 0.01% sodium and 0.03% salt, Suma brand, UK); baking powder (3% sodium, Royal brand, US); and whole milk (4.7% carbohydrate, 4.7% of which were sugars, 3.5% fat, 2.3% of which were saturated, 3.4% protein and 0.1% salt, Dunnes stores brand, UK). All food products were obtained from a local supermarket and stored under refrigerated or cool, dry conditions prior to sample preparation.

### 2.1. Reb A Concentration Adjustment

Optimised descriptive profiling (ODP) was used to determine the concentration of Reb A needed to replace the sweetness concentration of sucrose, ensuring iso-sweetness. These intensity tests were carried out twice using 21 assessors. Concentration adjustments for Reb A were carried out according to the method of [28,29,30,31] using the same concentrations of stevia (0.06–0.16 g/L) and 24 g/L sucrose. In order to compare the means of the obtained date, one-way ANOVA was used. Tukey’s post hoc test was used to adjust for multiple comparisons between treatment means using SPSS statistics 20 software (IBM, Armonk, NY, USA).

### 2.2. Sponge Cake Treatments

The formulation used for the preparation of the control sponge cake treatment was based on conversations had with local bakeries, cookbook recipes and associated websites. Three separate batches of sponge cake for all experimental treatments (8) were formulated and manufactured using recipes outlined in Table 1.

During the first phase of this trial, a control treatment with 0% SR was prepared. Sucrose was replaced by increments of 25% using a combination of inulin and Reb A (0, 25, 50 and 75%). Thus, 4 treatments were formulated during the first phase of this study and samples were identified as follows; SC0/0, SC25/0, SC50/0 and SC75/0. The most accepted sample was chosen from sensory acceptance testing and used for further analysis in determining the minimum levels of fat needed to maintain sensory properties associated with liking and OA of sugar-replaced sponge cake. Thus, four more formulations were prepared where fat was replaced sequentially by increments of 25% using pureed butter beans. The samples were identified as follows; SC30/0, SC30/25, SC30/50 and SC30/75, where the first digit denotes the SR replacement level and the second digit represents the FR level.

### 2.3. Sponge Cake Preparation

Butter was creamed in an electronic mixer (Kitchen Aid Professional mixer) and sugar was added gradually until soft and light in colour, at speed 2–3 for 4 min. Eggs (average and similar in size, without weighing) were added one at a time and beaten well between each addition at speed 2 for approximately 1 min, for each egg. Flour and baking powder were sifted in to the mixture and were mixed together at minimum speed for 2 min, before milk was added. For SR, inulin and stevia were added to the mixture during the creaming stage in partial replacement of sucrose. For FR, butter beans were firstly drained and then pureed in a Stephan mixer (UMC-5 Stephan u. Sohner & Co, Hameln, Germany) at 21 RPM for 5 min before being used in partial replacement of butter. Batter was poured into circular baking tins (9 inch) and cooked in a Zanussi convection oven (C. Batassi, Conegliano, Italy) for 30 min at 180 °C. After cooking, sponge cakes were left for 20 min in the tin. Then, cakes were placed on a rack to cool down to room temperature and were placed inside plastic storage containers before testing.

### 2.4. Sensory Analysis

#### 2.4.1. Sensory Acceptance Testing (SAT)

SAT was carried out in the sensory science laboratory at University College Cork according to ISO 11136:2014 [32]. Using 25 untrained assessors (*n* = 25) who were all familiar with the products being tested, SAT took place over six separate sessions as three independent trials were carried out for both phases of this study (FR and SR). A three-digit random code was used to assign the samples (20 × 20 × 20 mm). Sensory sessions were carried out at room temperature under white light. Participants were instructed to use the water provided to cleanse their palates between tastings and used the following hedonic descriptors to rate their degree of liking of sponge cake samples: appearance, flavour, texture, colour and aroma liking. Assessors were asked to indicate their degree of liking for samples on a 9-point, numbers-only hedonic scale. Words were only used to anchor the scale at both ends with the term “extremely dislike” on the far-left end of the scale and the term “extremely like” on the far-right-hand side of the scale. Overall acceptability (OA) of samples was also determined using this scale. The 150 responses were collected for each sample (25 + 25+ 25 × 2) in three independent trials for each phase, using 25 untrained assessors (samples were presented in duplicate). The statistical analysis method is shown in Section 2.7.

#### 2.4.2. Optimised Descriptive Profiling (ODP)

Optimised descriptive profiling [30,31,32] was also carried out in the special sensory science laboratory of University College Cork. A separate panel of 21 assessors (*n* = 21) who were all regular consumers of sponge cake and who had all received prior training in descriptive analysis were trained and participated in this separate ODP test. In order to prevent the carry-over effect and first order, these assessors were served all samples randomly [33]. ODP only took place on samples during the second phase of this study after an accepted minimum level of sucrose content was determined by SAT. Thus, ODP sessions ran concurrently with SAT sessions during fat optimisation and therefore took place over three weeks. Sensory descriptors were selected from the panel discussion as the most appropriate and reflected the main variation in the samples profiled. The consensus list of intensity descriptors (Table 2) was measured on a 10 cm continuous line scale. The term “none” was used as the anchor point for the 0 cm end of the scale and the term “extreme” was used as the anchor point for the 10 cm end of the scale, unless stated otherwise in Table 2. The samples (20 × 20 × 20 mm) were served coded and presented simultaneously to assessors [34]. All samples were measured on the same scale for each intensity descriptor and presented in replicate. The statistical analysis method is shown in Section 2.7.

### 2.5. Physicochemical Analysis

Physicochemical properties were determined for the following samples: SC0/0, SC30/0, SC30/25, SC30/50 and SC30/75.

#### 2.5.1. Texture Profile Analysis (TPA)

Two sponge cake samples (30 × 30 mm) were obtained from the centre of each cake and used for texture analysis. Thus, results obtained for TPA on the Texture Analyser 16 TA-XT2I (Stable Micro Systems, Surrey, UK) represent a mean of 6 values (3 × 2 = 6). Sponge cake samples were sliced horizontally to a height of 30 mm. A double compression test for 50% was used for TPA with use of a 35 mm (in diameter) flat-ended cylindrical probe (P/35), at a speed of 1 mm/s with a 5 s time between the two cycles. This was carried out in accordance with the method of [35]. The test was carried out in triplicate for all treatments.

#### 2.5.2. Colour

Two sponge cake samples (30 × 30 mm) from the centre of each cake were used for colour analysis. The CIE L*a*b* method was used for crust and crumb colour measurement. A Minolta CR-200B Chroma Meter (Minolta Camera Co. Ltd., Osaka, Japan) was used for L* (lightness), a* (green-red) and b* (blue-yellow) colour components measurement. Colour parameters were measured at two separate points directly from the top of each individual sponge cake sample. For this, the sponge cake samples were horizontally cut to remove the crust, and crumb colour was measured directly at two separate points. As two measurements for crust and crumb colour were taken for each individual sample and two samples were tested for each individual trial, of which there were three, crust and crumb colour values represent a mean of twelve measurements (2 × 2 × 3).

#### 2.5.3. Moisture and Fat

Moisture and fat were detected according to Bostian et al. [36]. Two samples (30 × 30 mm) from each cake treatment were used for moisture and fat determination. As three independent trials were carried out, results obtained for moisture and fat represent a mean of 6 values (3 × 2 = 6). The Büchi Mixer B-400 (Büchi Labortechnik AG, Switzerland) was used in order to homogenise for compositional analysis. Moisture content was detected on the CEM SMART system and fat content was detected on the SMART Trac system (CEM GmbH, kamp-Lintfort, Germany). Two fibreglass pads were placed in the drying chamber of the CEM SMART system and their weight was tared. The pads were removed and the homogenised samples (2–4 g) were weighed accurately on the pads using separate weighing scales. Following this, one pad was placed over the sample, pressed together and placed back into the drying chamber to begin drying. The moisture (%) was displayed within a few minutes. A sheet of Smart Trac film was used to determine a percent of fats by wrapping the fibreglass pads with the sample. After wrapping, the samples were placed in a tube of the Smart Trac system and positioned in the Smart Trac NMR unit. After about 5 min, the percentage of fat was determined.

#### 2.5.4. Protein

Results obtained from protein analysis represent a mean of 6 values (3 × 2 = 6). Protein content (%) was determined using the Kjeldahl method which was carried out according to [37]. Before testing commenced, the digestion block (Foss Tecator Digestor, Hillerød, Denmark) was pre-heated to 410 °C. Samples (0.5–2.0 g) were weighed accurately into digestion tubes. Two “kjeltabs” were added to each sample in the fume hood followed by 15 mL of sulphuric acid and 10 mL hydrogen peroxide. Two controls containing no sample were prepared in the same way. Tubes were placed in the heated digestion block and left there for roughly 30–40 min until they became colourless. At this point, the distillation unit was turned on and rinsed out by connecting a blank tube and receiver flask to the unit and pressing the steam button. Distilled water (50 mL) was added to each digested sample in the fume hood after the samples had cooled thoroughly. Samples were placed one by one into the distillation unit (Foss Kjeltec 2100, Hillerød, Denmark) along with a receiver flask containing 50 mL of 4% boric acid with an indicator. After the distillation process was complete, the contents of the receiver flask were titrated with 0.1 N hydrochloric acid until the green colour reverted back to the original red colour. Nitrogen content was converted to protein content using the factor 6.25.

#### 2.5.5. Ash

Results obtained represent a mean of 6 values (3 × 2 = 6). A muffle furnace was used for ash content (%) detection. Homogenised samples (5 g) were weighed into small silica dishes and placed into the muffle furnace. The samples were heated up to 600 °C until only the inorganic material was left which was indicated by the light-white colour of the samples. The silica dishes containing the samples were then placed in a desiccator to cool and the dishes were weighed carefully. Ash (%) was calculated using the following equation:% Ash = weight of Ash × 100 weight of sample(1)

#### 2.5.6. Carbohydrate

Carbohydrate (%) content was determined using the following calculation:% Carbohydrate = 100 − (Moisture% + Fat% + Protein% + Ash%)(2)

#### 2.5.7. Dietary Fibre

Total dietary fibre was established according to the AOAC 985.29 method [38]. Sponge cake samples went through sequential enzymatic digestion using heat-stable α-amylase (95–100 °C, pH 6.0, 15 min), Subtilisin A (60 °C, pH 7.5, 30 min) and amyloglucosidase (60 °C, pH 4.5, 30 min) to remove digestible carbohydrates and protein. To establish total dietary fibre content, the enzyme digests were precipitated with four volumes of 95% (*v*/*v*) ethanol, filtered and then washed with 78% (*v/v*) ethanol and acetone before being dried in an oven at 110 °C.

#### 2.5.8. Total Sugars and Sucrose Content

Total sugars and sucrose content of samples was determined by ion chromatography, using a method developed internally by an independent accredited laboratory based in Ireland.

### 2.6. Sponge Cake Images

Photographs of the following sponge cake samples: SC0/0, SC30/0, SC30/25, SC30/50 and SC30/75, were taken one day after baking in the photograph room of the Food Science Building, University College Cork. Photographs were taken using a digital camera, Nikon D5300, equipped with a lens, Nikon DX AF-P NIKKON (18–55 mm, 1:3.5−5.6 G), Japan. Images were taken without flash with the following modes: macro, F6.3, 1/125, ISO 400. After images were taken, sharpness was increased to 100% and the saturation was adjusted for all images.

### 2.7. Statistical Analysis

Raw data obtained from sensory and physicochemical analysis were coded in Microsoft Excel. For sucrose optimisation, the significance of hedonic sensory properties in discriminating between the samples was analysed using ANOVA and Tukey’s post hoc test (SPSS statistics 20 software (IBM, Armonk, NY, USA). The relationship between the set of samples (4) and the set of hedonic sensory variables was determined by partial least squares (PLS) regression using Unscrambler software (Unscrambler 10.3 CAMO software ASA, Trondheim, Norway). In the PLS regression, only hedonic sensory properties that discriminated significantly between samples were used. The X-matrix was defined as the different sample treatments. The Y-matrix contained the significant sensory variables of the design. For fat optimisation, the relationship between the set of sample treatments (X) and the set of sensory (hedonic and intensity) and physicochemical variables (Y) was examined by PLS regression. Again, only sensory and physicochemical properties that discriminated significantly between samples (4) were used. Both the sensory and physicochemical data were normalised during pre-processing by taking the logarithm to achieve uniform precision over the whole range of variation. Data were also standardised by dividing each variable (sensory and physicochemical) by its standard deviation. This process was necessary as the units of the studied variables were different. To achieve significant results for the relationships determined in quantitative PLS regression analysis, coefficients were analysed by jack-knifing which was based on custom cross-validation and stability plots [39]. Statistical significance for the relationships analysed by PLS was defined as *p* < 0.05–0.01 (significant), *p* < 0.01–0.001 (highly significant) and *p* < 0.001 (extremely significant).

TPA and proximate composition data were presented as a mean of six values ± standard deviation. Estimated fibre and sucrose contents were presented as a mean of three values ± standard deviation. Colour (crust and crumb) data were presented as a mean of twelve values ± standard deviation. One-way ANOVA was used to compare the means of the data obtained from physicochemical analysis. Tukey’s post hoc test was used to adjust for multiple comparisons between treatment means using SPSS statistics 20 software (IBM, Armonk, NY, USA).

## 3. Results and Discussion

### 3.1. Reb A for Sucrose Replacement

The sweetness rankings of six different concentrations of Reb A and one standard solution of sucrose are presented in Table 3.

The Reb A solution containing 0.069 g/L did not obtain significantly different scores from the standard sucrose solution with regard to sweetness intensity. For this reason, a sucrose-to-Reb A ratio of 1:350 was chosen. This means that for samples containing 25% SR, which equates to a reduction of 58.3 g sucrose in the formulation, 0.17 g of Reb A was used to replace the sweetness (58.3/350). The same method was applied to samples with 50 and 75% SR. Inulin was added on a weight by weight basis.

### 3.2. Sensory Analysis

#### 3.2.1. Sensory Acceptance of Sucrose-Replaced Cakes

The relationship between hedonic sensory variables (Y) and sponge cake samples prepared with increasing levels of SR (X) is visually represented by a partial least squares regression plot (PLSR) shown in (Figure 3).

Most of the variation is shown in Factor-1, where 33% of the data in X explain 56% of the Y data. The SC0/0 sample which is situated in the outer circle of the lower right quadrant made a significant contribution to Factor-1. This sample was highly correlated with all liking parameters (aroma, appearance, colour, texture and flavour) and OA. The SC25/0 sample, which is positioned in the outer circle of the upper right quadrant, was also positively associated with all liking terms and OA. The SC75/0 sample, which is situated in the outer circle of the upper left quadrant, was negatively correlated with all liking parameters and OA. The SC/50 sample, which is positioned in the outer circle of the bottom left quadrant, made a significant contribution to Factor-2 and was slightly negatively correlated with liking terms and OA. It is evident from the plot that all liking parameters were correlated as shown by their proximity to each other, particularly so for texture liking and OA. To aid further understanding of the relationship between sensory terms and sponge cake samples, the significance of the estimated regression coefficients for the relationship between these two sets of variables can be seen in Table 4.

Resembling results which are visually represented in the PLSR plot, the SC75/0 sample was significantly negatively associated with aroma, colour, flavour liking and OA (*p* < 0.05) and significantly negatively associated with texture liking (*p* < 0.01). Although it was evident from the PLS plot (Figure 3) that a 50% replacement of sucrose with inulin and Reb A was unsuccessful in terms of liking, the negative response was only significant for texture (*p* < 0.01) and subsequently OA (*p* < 0.05) (Table 4). In a study conducted on orange cakes, the addition of inulin negatively affected texture liking, whereas flavour liking and aroma liking were not affected [40]. In a study conducted by [17], muffins containing 30% SR with inulin and Reb A did not differ from the reference sample with regard to aromatics, browning, cracks on crust or buttery and sweet flavour. Texture liking was important to the acceptability of samples in this study; therefore, low texture liking scores were a good predictor for the rejection of samples. The liking of aroma, appearance, colour and flavour was not significantly affected by up to a level of 50% SR (Table 4). As samples containing 25% SR were so similar to the reference sample in this study and because a 30% replacement was achieved by [17] without significantly affecting important sensory attributes, an SR level of 30% was chosen for further experiments to determine the minimum levels of fat needed to maintain sensory properties associated with the liking and OA of samples.

#### 3.2.2. Sensory and Physicochemical Properties of Reduced Sucrose Cakes with Increasing Levels of FR

The second part of this study involved the sequential replacement of fat in 30% sucrose-replaced sponge Cake with Pbb. The relationship between sensory and physicochemical variables (Y) and sucrose-replaced sponge cake prepared with 0, 25, 50 and 75% FR with Pbb (X) is visually represented by a PLSR plot in Figure 4.

The following hedonic sensory terms were left out of PLSR analysis because they did not significantly discriminate between samples: appearance, colour and texture liking. The following intensity sensory terms were omitted for the same reason: porous appearance, crumbly texture and dense texture. The following physical parameters and compositional properties were also left out of PLSR analysis for the same reason: hardness, gumminess and chewiness (N), springiness (mm) and cohesiveness, crumb a* and b* values and ash, carbohydrate, total sugars and sucrose content. Most of the variation in the plot is shown in Factor-1, where 33% of the X data explain 51% of the data in Y. The SC30/0 sample, which is shown in the outer circle of the lower right quadrant, makes a significant contribution to Factor-1. It is evident that the SC30/0 sample was positively associated with aroma and flavour liking and OA which can also be seen on the right-hand side of the plot. The following intensity sensory terms were correlated with liking parameters and OA and can be seen in close proximity with these hedonic parameters on the right-hand side of the plot: sweet, buttery and caramel aroma, sweet taste, butter flavour, crust and crumb darkness, moist texture, springiness and fat content (%). These intensity sensory terms can be considered as positive sensory variables associated with liking and OA. Actual fat content (%) was a big predictor of acceptability as it can be seen in very close proximity with aroma and flavour liking and OA. The SC30/75 sample, which is positioned in the outer circle of the lower left quadrant, was highly negatively correlated with liking parameters (aroma and flavour), OA and all sensory properties associated with liking and OA, situated on the right-hand side of the plot. This sample was correlated with intensity sensory terms and physicochemical variables situated on the left-hand side of the plot (perceived texture hardness, crust lightness, redness and yellowness and actual moisture, protein and fibre content (%)). These parameters were therefore negatively correlated with liking parameters and OA.

The significance of the estimated regression coefficients for the relationship between samples and sensory and physicochemical parameters can be seen in Appendix A (Appendix A). After custom cross-validation of PLSR analysis, the sample containing 75% replacement of fat with Pbb (SC30/75) was the only sample significantly negatively associated with aroma and flavour liking (*p* < 0.05) and significantly negatively associated with OA (*p* < 0.01). Pureed butter beans were therefore successful in terms of acceptance and liking in the replacement of fat in sucrose-replaced sponge cake up to a level of 50%. Similar results were reported by [27], who found that significant differences existed only when 75% of shortening was replaced with pureed cannellini beans in brownies.

The SC30/75 sample was significantly negatively associated with all intensity attributes associated with liking (*p* < 0.05), particularly sweet taste and butter flavour (*p* < 0.01) (Appendix A, Appendix A). A reduction in perceived butter flavour was expected at this level of butter replacement with Pbb and similar results on FR in biscuits have been reported by Laguna et al. [41]. The combined effect of fat and sugar on sensory acceptability has been demonstrated previously by Biguzzi et al. [42], who found that perceived sweetness intensity declined with fat reduction in biscuits. Therefore, it is not surprising that sweetness intensity was significantly affected at a level of 75% FR in this study. A reduction in aroma attributes such as butter, caramel and sweet is also not surprising. Butter aroma is stronger than the neutral aroma of butter beans, and a reduction in perceived butter aroma may have had a carry-over effect on caramel and sweet aroma perception, considering the synergistic relationship between sugar and butter in cake products. The decrease in perceived moist texture was also expected at this level of FR as fats have a lubricating effect and produce a sensation of moistness in the mouth [43]. Contradictory to sensory results obtained for moist texture perception, the SC30/75 sample was significantly positively (*p* < 0.05) associated with actual moisture content (%) (Appendix A). The difference in results obtained for actual moisture content (%) and sensory results obtained for moist texture perception (Figure 3) highlights the importance of fat/butter content to the perception of moist texture, lubricity and, subsequently, acceptance. The moisture content of samples will be discussed in further detail in Section 3.4.

As mentioned, results obtained for texture profile analysis (TPA) did not discriminate between samples and so these parameters were omitted from the PLS plot. TPA results will be discussed in further detail in Section 3.3. Sensory results obtained for perceived sample hardness, however, did discriminate between samples and the SC30/75 sample was significantly associated with this intensity attribute (*p* < 0.05). Perceived hard texture was negatively correlated with positive sensory properties and liking parameters; however, it was not negatively correlated with texture liking as liking of texture did not discriminate between any samples. As texture liking did not discriminate between samples, it can be said that perceived sample hardness was not a determinate for the lower acceptance scores obtained by the SC75/75 sample.

Crust lightness (L*), which is positioned on the left-hand side of the plot (Figure 3), was significantly positively associated with the SC30/75 sample (*p* < 0.05) (Appendix A). This sample was significantly negatively associated with the sensory terms crust darkness and crumb darkness (Appendix A), which were highly correlated with one another (Figure 3). Crust darkness and crumb darkness were intensity attributes most associated with samples containing low levels of FR (Figure 3). Confortiv et al. [20] also reported an increase in lightness of biscuit samples with FR using carbohydrate-based fat replacers. Samples containing 75% FR were also more yellow in colour than any other samples (*p* < 0.05) (Appendix A). Instrumental results obtained for colour will be discussed in further detail in Section 3.3. Although perceived colour intensity and instrumental colour parameters discriminated between samples, as mentioned, liking of colour and appearance did not discriminate and, as a result, these liking parameters were omitted from the PLS plot. This finding suggests that in addition to textural parameters, colour parameters were not extremely important to the OA of samples. Contradictory to our findings on colour liking, in a study conducted on oatmeal cookies, samples containing higher levels of FR (50 and 75) with white beans were significantly different to samples containing 0 and 25% FR with regard to colour liking [27].

As flavour liking and aroma liking were the only hedonic parameters that significantly discriminated between samples, flavour (butter, sweet) and aroma (butter, sweet and caramel) intensity attributes were, therefore, determinants of OA. Although a dryer and harder texture was associated with samples containing 75% FR (Appendix A, Appendix A), these textural parameters were not found to significantly affect the texture liking of samples. Samples containing this level of FR were perceived as lighter and more yellow in colour, which was supported by instrumental analysis.

However, colour or appearance liking did not significantly discriminate between sample treatments. These findings demonstrate that Pbb were successful in replacing fat in terms of colour and texture liking but were unsuccessful in terms of flavour and aroma liking up to a level of 75% replacement in 30% sucrose-replaced sponge cake. Hence, aroma liking and flavour liking were determinants of OA.

No significant differences were found in relation to any intensity sensory attributes or hedonic parameters up to a level of 50% FR in sucrose-replaced sponge cake. Thus, flavour liking and aroma liking and, subsequently, OA were unaffected by this level of FR.

### 3.3. Physical Properties and Images of Sponge Cake Samples

Mean values from physical analysis on the following samples of sponge cake are displayed in Table 5: SC0/0, SC30/0, SC30/25, SC30/50 and SC30/75. Cross-section images of sponge cake samples are depicted in Figure 5.

A 30% SR in sponge cake significantly increased sample hardness (*p* < 0.05). This result was expected as sucrose plays a vital role in the tenderisation of baked goods, and similar results in relation to increased sample hardness with the addition of inulin have been reported by O’ Brien et al. [44] and Volpini-Rapina et al. [40] in breadcrumbs and orange cakes, respectively. The chewiness (N) of samples also decreased significantly after a 30% SR. As previously mentioned, instrumental texture hardness values (N) did not significantly discriminate between samples with increasing levels of FR using Pbb (Table 5). As fats also play a vital role in the tenderisation of dough, an increase in crumb hardness was expected at higher levels of replacement. Souza et al. [45] reported an increase in the crumb firmness of pound cakes with only a 25% replacement of butter with green banana puree. However, as mentioned, instrumental TPA values did not correlate with sensory results and samples containing 75% FR were perceived as harder. This sample was also perceived as significantly dryer which may have had a carry-over effect on the perceived hardness of samples.

A 30% SR in sponge cake using inulin and Reb A did not significantly affect instrumental crust or crumb colour properties. This finding was supported by images depicted in Figure 5. Crust lightness, however, increased as FR increased in samples containing 30% SR in the range of (40.5 ± 0.41) for the SC30/0 sample and (49.8 ± 1.32) for the SC30/75 sample (*p* < 0.05) (Table 5). This finding was also visually supported by the images depicted in Figure 5. Crust yellowness (b* values) increased as FR increased in sucrose-replaced samples in the range of (15.8 ± 0.38) for the SC30/0 sample and (17.5 ± 0.78) for the SC30/75 sample (*p* < 0.05). Instrumental results obtained for crust colour therefore correlate with sensory results. The increase in crust yellowness observed with increasing levels of FR in our study could be attributed to the yellow colour associated with butter beans.

Crumb lightness values (b*) also discriminated between samples with increasing levels of FR; however, uneven trends were observed with the SC30/25 and the SC30/50 samples obtaining the highest values for crumb lightness (*p* < 0.05). This result did not correlate with sensory results as the SC30/75 sample was the sample most negatively correlated with crumb darkness, which means they were perceived as the lightest samples. A plausible reason for this is that sensory panellists failed to discriminate between crust and crumb colour, and this is supported by the high correlation between these two attributes, depicted in Figure 2. Images displayed in Figure 3 may help to explain the difficulty to distinguish between crust and crumb colour.

### 3.4. Proximate Composition of Sponge Cake Samples

As displayed in Table 6, a 30% SR in sponge cake did not significantly affect the moisture content (%) of samples. Moisture content, however, increased significantly as FR increased in 30% sucrose-replaced samples in the range of 22.1% for the SC30/0 sample to 32.4% for the SC30/75 sample (*p* < 0.05) (Table 6).

Butter beans contain a significantly higher percentage of moisture (75%) compared to the butter used in this trial (15%), which can explain the increase in moisture with the substitution of butter for Pbb. As mentioned, sensory results obtained for moist texture did not correlate with instrumental results, which highlights the importance of butter content to the perception of lubricity and moist texture. Fat content remained the same after a 30% SR as expected, but it decreased significantly with each level of FR in the range of 12.6% for the SC30/0 sample to 4.5% for the SC30/75 sample (*p* < 0.05). As a 50% FR was permitted in sucrose-replaced sponge cake samples in relation to sensory acceptance, this equated to a total fat reduction of 42%. According to the standards set by the Food Safety Authority (2016) [46], this reduction in fat content permits the claim “reduced fat”. A 30% SR did not affect the protein content of samples. Trends show that protein content increased with each level of FR; however, a significant increase was only observed for the sample containing 75% FR (*p* < 0.05). The increase in protein content can be attributed to the protein content of Pbb. Carbohydrate content decreased significantly after a 30% replacement of sucrose for inulin and Reb A, as expected (*p* < 0.05). Carbohydrate content remained constant in samples with increasing levels of FR at roughly 55%. Sucrose content and total sugars also decreased significantly after a 30% SR, as expected (*p* < 0.05). Similar to results obtained for carbohydrate content, sucrose content and total sugars remained constant with increasing levels of FR, as expected. Although carbohydrate content was high in all samples with increasing levels of FR (55%), less than half of the carbohydrate was in the form of sucrose and total sugars. As a 30% SR was achieved in terms of sensory acceptance, this equated to a 28% reduction in total sugars. Dietary fibre increased significantly with the substitution of 30% sucrose for inulin and Reb A (*p* < 0.05), which can be attributed to the fibre present in inulin. Trends show that fibre content increased with each level of FR but only significantly so after a 50% FR with Pbb (*p* < 0.05). The optimised sample (SC30/50) contained 2.8 g/100 g of dietary fibre which was a total increase of 115% from the original sample, and a total energy reduction of 15% was achieved.

## 4. Conclusions

Texture liking and, subsequently, OA were negatively affected by a 50% SR using a combination of inulin and Reb A. However, other important hedonic attributes such as flavour liking and aroma liking were not negatively affected by this level of replacement. This demonstrates the value of these replacers in relation to maintaining flavour properties associated with liking. Further work to improve texture, such as the addition of emulsifiers, for example, polysorbates, or the addition of hydrocolloids, is necessary to permit higher levels of sucrose replacement using this combination of replacers.

Pureed butter beans were very successful in the replacement of fat by up to a level of 50% replacement. Flavour and aroma intensity attributes and, subsequently, liking and OA were negatively affected by a 75% FR. Perhaps the addition of flavourings to compensate for the reduction in flavour lost with butter would permit higher levels of fat replacement using pureed butter beans.

The present study achieved a simultaneous 50% FR in 30% sucrose-replaced sponge cake, using natural substitutes and without negatively affecting OA. Sponge cake samples contained 42% less fat and 28% less sucrose than the original sample. Thus, the criteria to meet the standard for a “reduced fat” claim were achieved. More work is necessary to permit further sucrose and fat replacement using these ingredients and without negatively affecting important quality parameters, in order to meet the criteria for reduced/low-sugar and low-fat health claims. Optimised samples contained 115% more dietary fibre than the original sample.

## Figures and Tables

**Figure 1 foods-10-00254-f001:**
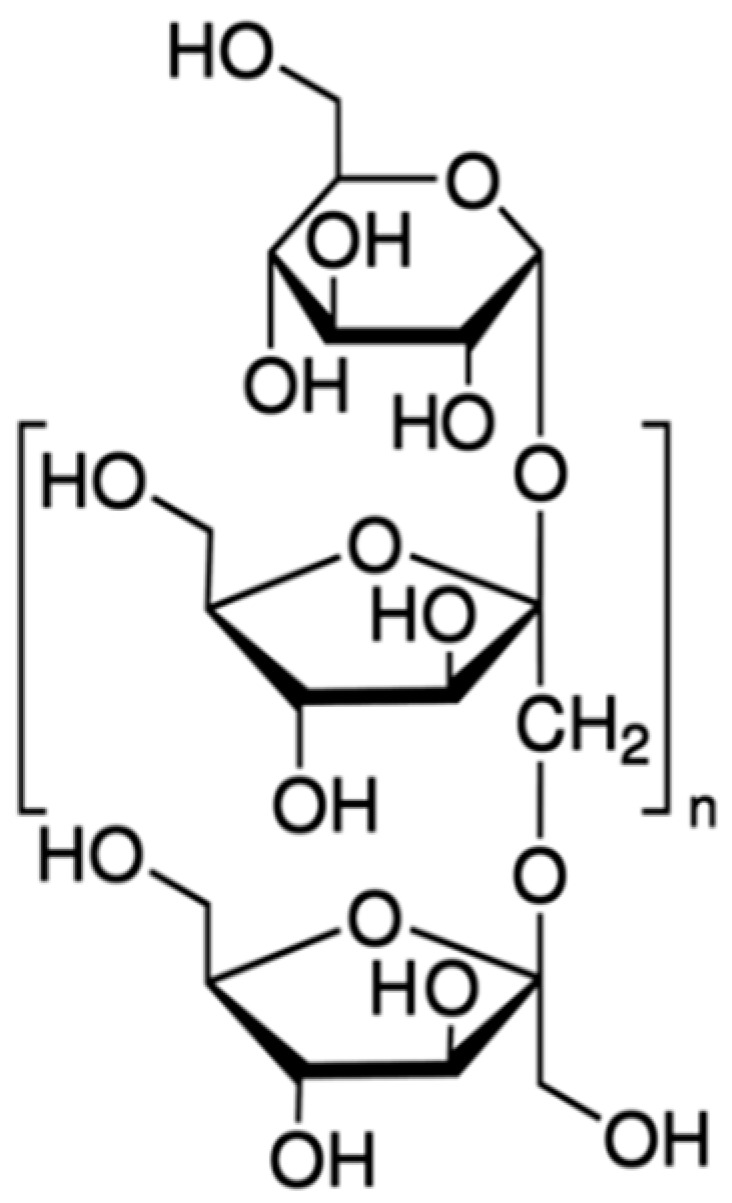
Chemical structure of inulin.

**Figure 2 foods-10-00254-f002:**
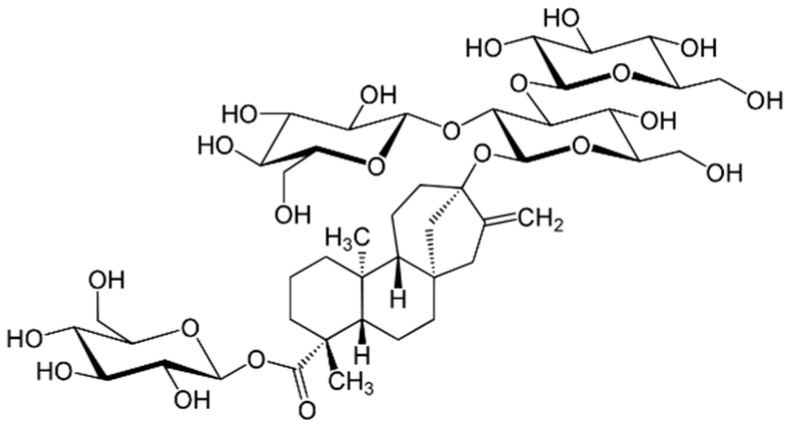
Structure of Rebaudioside A.

**Figure 3 foods-10-00254-f003:**
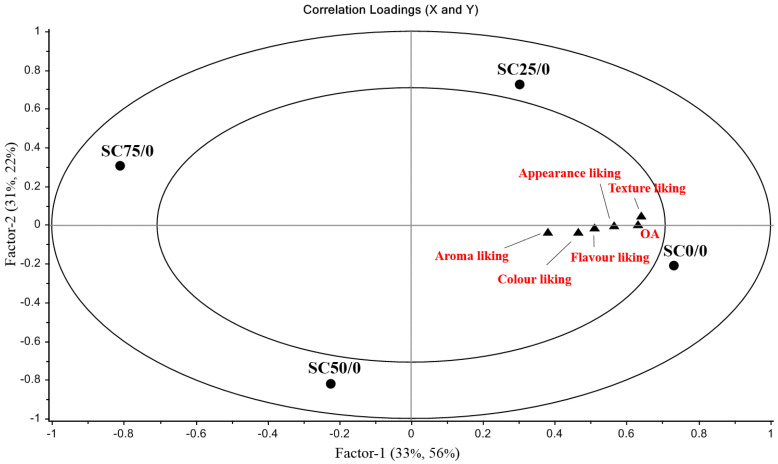
Partial least squares regression (PLSR) plot for the relationship between sponge cake samples with increasing levels of sucrose replacement: 0, 25, 50 and 75%, and hedonic sensory variables.

**Figure 4 foods-10-00254-f004:**
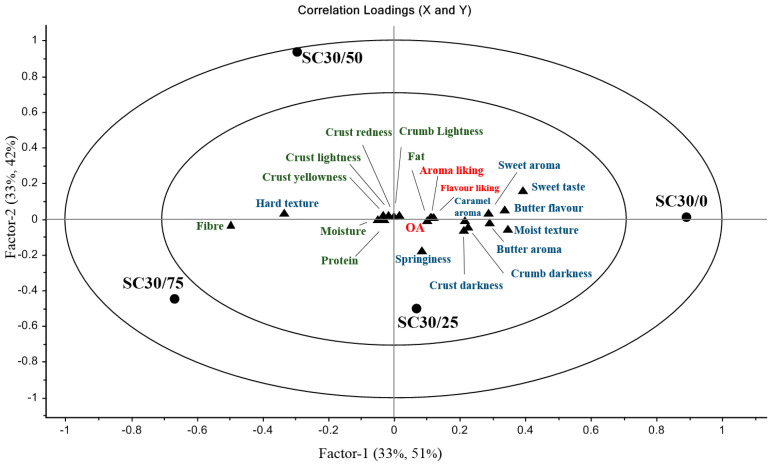
Partial least squares regression (PLSR) plot for the relationship between reduced sugar sponge cake (30%) prepared with increasing levels of fat replacement: 0, 25, 50 and 75%, sensory and physicochemical variables, hedonic and intensity sensory terms and physicochemical parameters.

**Figure 5 foods-10-00254-f005:**
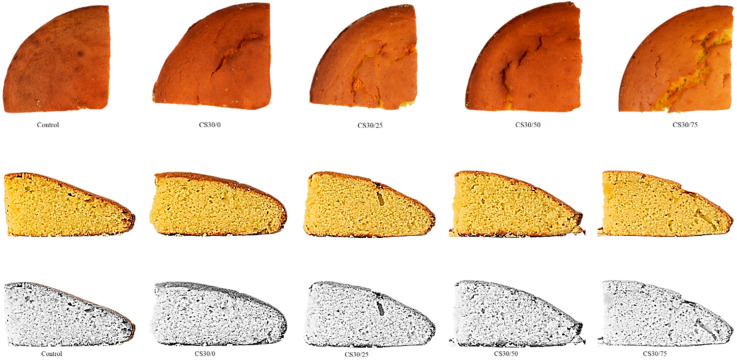
Top view (crust) and cross-section images of sponge cake fat reduced samples. From the left: control, S30/0, S30/25, S30/30, S30/50, S30/75.

**Table 1 foods-10-00254-t001:** Formulation of different sponge cake treatments.

		%								
	Samples	Sucrose	Inulin	Reb A	Butter	Pureed Butter Beans	Flour	Milk	Baking Powder	Eggs
Sucrose replacement	SC0/0	26.1	0	0	18.6	0	26.1	1.7	0.7	26.8
	SC25/0	19.55	6.49	0.02	18.6	0	26.1	1.7	0.7	26.8
	SC50/0	13.04	13.0	0.04	18.6	0	26.1	1.7	0.7	26.8
	SC75/0	6.51	19.49	0.06	18.6	0	26.1	1.7	0.7	26.8
Fat replacement	SC30/0	18.23	7.82	0.02	18.6	0	26.1	1.7	0.7	26.8
	SC30/25	18.23	7.82	0.02	13.96	4.7	26.1	1.7	0.7	26.8
	SC30/50	18.23	7.82	0.02	9.3	9.3	26.1	1.7	0.7	26.8
	SC30/75	18.23	7.82	0.02	4.7	13.96	26.1	1.7	0.7	26.8

SC; sponge cake, the first digit represents the sucrose replacement level and the second digit represents the fat replacement level of samples.

**Table 2 foods-10-00254-t002:** Consensus list of sensory descriptors and definitions selected by panel and used in ranking descriptive analysis of sponge cake.

Attributes	Anchor Points on Scale	Definition
Springiness	None–extreme	Rate⁄speed by visual observation that the sponge cake returns to its original shape after the deforming force is removed
Appearance		
Crust darkness	Light–dark	Degree of colour darkness ranging from light brown to dark brown
Porous	None–extreme	Amount of bubbles and voids present in the inner mass of the sponge cake
Aroma		
Sweet aroma	None–extreme	Fundamental smell sensation of which sucrose is typical
Buttery aroma	None–extreme	Aromatics associated with butter produced from cow’s milk
Caramel aroma	None–extreme	Odour produced when caramelising sugar without burning it
Texture		
Moisture	Dry–moist	Wet texture in mouth
Hardness	Soft–hard	The resistance of the cake to breaking upon pressure of the front teeth during biting
Crumbly	None–extreme	Easily broken up in the mouth into a lot of little pieces
Dense	Light–heavy	A heavy texture in mouth
Taste		
Sweet taste	None–extreme	Taste sensation associated with sucrose
Buttery flavour	None–extreme	Flavour sensation associated with butter; creamy mouthfeel and buttery aroma

**Table 3 foods-10-00254-t003:** Iso-sweetness of Rebaudioside A in aqueous solutions.

Sweetener	Concentration (g/L)	Mean Scores	Dilution Factor
Reb A	0.060	1.2 ^a^	400
Reb A	0.069	5.4 ^b^	350
Reb A	0.080	6.2 ^c^	300
Reb A	0.096	6.3 ^c^	250
Reb A	0.120	7.4 ^d^	200
Reb A	0.160	9.4 ^d^	150
Sucrose	24.0	5.6 ^b^	n/a

^abcd^ mean values (±standard deviation) in the same column bearing different superscripts are significantly different, *p* < 0.05. n/a: Only Reb A solution was deluted. Sucrose was used in solid state.

**Table 4 foods-10-00254-t004:** Significance of estimated regression coefficients (ANOVA values) for the relationship of hedonic sensory parameters (Y) and sponge cake samples prepared with increasing levels of sucrose replacement (X).

Hedonics						
	Aroma Liking	Appearance Liking	Colour Liking	Texture Liking	Flavour Liking	Overall Acceptability
SC0/0	0.57	0.68	0.56	0.57	0.68	0.56
SC25/0	0.55	0.84	0.56	0.57	0.76	0.66
SC50/0	0.57	0.84	0.57	−0.04 *	0.83	−0.03 *
SC75/0	−0.05	−0.03 *	−0.04 *	−0.01 **	−0.03 *	−0.02 *

Significance of regression coefficients * = *p* ≤ 0.05, ** = *p* ≤ 0.01, (−) indicates whether the relationship is negatively correlated.

**Table 5 foods-10-00254-t005:** Physical properties of sponge cake samples.

		SC0/0	SC30/0	SC30/25	SC30/50	SC30/75
TPA para-meters	Hardness (N)	8.2 ± 0.55 ^a^	15.1 ± 0.14 ^b^	13.2 ± 0.44 ^b^	14.1 ± 0.76 ^b^	14.2 ± 1.52 ^b^
	Gumminess (N)	4.8 ± 0.66 ^a^	6.0 ± 0.78 ^b^	5.8 ± 0.33 ^b^	5.9 ± 0.72 ^b^	6.4 ± 0.60 ^b^
	Chewiness (N)	8.3 ± 0.58 ^a^	3.9 ± 0.59 ^b^	3.9 ± 0.82 ^b^	3.8 ± 0.75 ^b^	3.7 ± 0.55 ^b^
	Springiness (mm)	0.7 ± 0.12 ^a^	0.6 ± 0.03 ^a^	0.7 ± 0.04 ^a^	0.6 ± 0.07 ^a^	0.7 ± 0.05 ^a^
	Cohesiveness	0.6 ± 0.71 ^a^	0.4 ± 0.04 ^a^	0.4 ± 0.03 ^a^	0.4 ± 0.04 ^a^	0.4 ± 0.08 ^a^
Crust colour	Lightness (L*)	39.8 ± 0.65 ^a^	40.5 ± 0.41 ^a^	41.0 ± 0.77 ^a^	45.5 ± 0.40 ^b^	49.8 ± 0.32 ^c^
	Redness (a*)	15.1 ± 0.25 ^a^	15.8 ± 0.38 ^a^	16.8 ± 0.88 ^ab^	16.9 ± 0.60 ^b^	17.5 ± 0.78 ^b^
	Yellowness (b*)	24.1 ± 0.61 ^a^	24.8 ± 0.88 ^a^	25.2 ± 0.66 ^ab^	29.8 ± 0.20 ^b^	34.1 ± 0.40 ^c^
Crumb colour	Lightness (L*)	69.2 ± 0.44 ^a^	70.8 ± 0.50 ^a^	71.6 ± 0.31 ^ab^	72.3 ± 0.45 ^b^	70.7 ± 0.51 ^a^
	Redness (a*)	−3.1 ± 0.32 ^a^	−2.8 ± 0.21 ^a^	−2.7 ± 0.42 ^a^	−2.8 ± 0.45 ^a^	−2.7 ± 0.36 ^a^
	Yellowness (b*)	31.0 ± 0.24 ^a^	29.8 ± 0.60 ^a^	27.7 ± 0.80 ^a^	27.2 ± 0.30 ^a^	28.0 ± 0.54 ^a^

^abc^ mean values (±standard deviation) in the same row bearing different superscripts are significantly different, *p* < 0.05.

**Table 6 foods-10-00254-t006:** Proximate composition (%) of sponge cake samples.

	SC0/0	SC30/0	SC30/25	SC30/50	SC30/75
Moisture	21.6 ± 0.55 ^a^	22.1 ± 0.01 ^a^	27.5 ± 0.03 ^b^	28.6 ± 0.42 ^b^	32.4 ± 0.06 ^c^
Fat	12.9 ± 0.41 ^a^	12.6 ± 0.74 ^a^	10.1 ± 0.26 ^b^	7.5 ± 0.35 ^c^	4.5 ± 0.33 ^d^
Protein	6.1 ± 0.47 ^a^	6.0 ± 0.75 ^a^	6.7 ± 1.40 ^ab^	6.8 ± 0.51 ^ab^	7.5 ± 0.33 ^b^
Ash	1.0 ± 0.35 ^a^	1.3 ± 0.06 ^a^	1.6 ± 0.63 ^a^	1.3 ± 0.18 ^a^	1.4 ± 0.05 ^a^
Carbohydrate	58.4 ± 0.28 ^a^	55.0 ± 0.60 ^b^	54.2 ± 0.25 ^b^	55.5 ± 0.10 ^b^	54.2 ± 0.55 ^b^
Sucrose	28.9 ± 0.66 ^a^	20.4 ± 0.55 ^b^	21.0 ± 0.55 ^b^	21.0 ± 0.54 ^b^	21.1 ± 0.41 ^b^
Total sugars	29.8 ± 0.52 ^a^	20.8 ± 0.41 ^b^	21.4 ± 0.33 ^b^	21.5 ± 0.58 ^b^	21.4 ± 0.65 ^b^
Dietary fibre	1.3 ± 0.57 ^a^	2.0 ± 0.71 ^b^	2.4 ± 0.22 ^bc^	2.8 ± 0.46 ^c^	3.3 ± 0.22 ^cd^
Energy (Kcal/100 g)	374	357	335	317	287

^abcd^ mean values (±standard deviation) in the same row bearing different superscripts are significantly different, *p* < 0.05.

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
