# Peer review of "The Application of Pureed Butter Beans and a Combination of Inulin and Rebaudioside A for the Replacement of Fat and Sucrose in Sponge Cake: Sensory and Physicochemical Analysis"

_foods, 2021, doi:10.3390/foods10020254_

Round 1
Reviewer 1 Report
A well written paper. Authors should please address the points.
Line 60: Change “preform” to “perform”
Line 78-81: Authors should include more literature on Reb A and why this was chosen besides the fact that other authors have used Reb A. Example calorie level? Advantages? Sweetness equivalent compared to sucrose?
Line 80-81: Why are steviol glycosides such as Reb A considered natural?
Line 169: Are sponge cakes the same as chocolate brownies?
Author Response
Dear Reviewer 1,
Thank you for the careful consideration of our paper. We made all changes required and marked them in red in the text.
Line 60: Change “preform” to “perform”.
- changed.
Line 78-81: Authors should include more literature on Reb A and why this was chosen besides the fact that other authors have used Reb A. Example calorie level? Advantages? Sweetness equivalent compared to sucrose?
- In the text we added a chemical formula of Inulin and Reb. A (new Figs. 1, 2) and the following information:
Reb. A is the steviol glycoside (Fig. 2) which sweetness 300 time greater in comparison with a sucrose [18, 19]. Reb. A represents white odorless crystals. Steviol glycosides such as Reb A. were approved by the European Union for use in foods and are considered natural intense sweeteners [20]. These sweeteners are a special class of intense sweeteners as they are considered natural unlike most other intense sweeteners [20] as they are produced from Stevia leaves [21].
4 new Refs added:
Kinghorn A.D.; Wu C. D., Soejarto D. D; Compadre C. M., Stevioside. In Alternative Sweeteners, 3rd ed.; Nabors L. O., Ed.; Dekker Marcel, Inc: New York, USA, 2001; pp 167–183.
Jorge K. (2003). SOFT DRINKS Chemical Composition. In Encyclopedia of Food Sciences and Nutrition (Second Edition), pp. 5346-5352. https://doi.org/10.1016/B0-12-227055-X/01101-9
Izawa, Kunisuke; Amino, Yusuke; Kohmura, Masanori; Ueda, Yoichi; Kuroda, Motonaka (2010). "4.16 - Human–Environment Interactions – Taste". In Liu, Hung-Wen (Ben); Mander, Lew (eds.). Comprehensive Natural Products II. 4. Elsevier. pp. 631–671.
Li Yan, Li Yangyang, Wang Yu , Chen Liangliang, Yan Ming, Chen Kequan, Xu Lin, Ouyang Pingkai (2016). Production of Rebaudioside A from Stevioside Catalyzed by the Engineered Saccharomyces cerevisiae. Appl. Biochem Biotechnol, 178(8), 1586-98. https://10.1007/s12010-015-1969-4.
Line 80-81: Why are steviol glycosides such as Reb A considered natural?
- "Stevioside and rebaudioside A are steviol glycosides (SG) from Stevia rebaudiana The extracted products are natural sweeteners that are up to 300 times sweeter than sucrose and show a sufficient thermal stability, making them suitable as sucrose replacers in sweet bakery products" [20].
The link is provided to the paper in Reference in the manuscript: https://ifst.onlinelibrary.wiley.com/doi/full/10.1111/ijfs.12617
In the text was added: These sweeteners are a special class of intense sweeteners as they are considered natural unlike most other intense sweeteners [21 new] as they are produced from Stevia leaves [20].
A new link added in Ref.
Line 169: Are sponge cakes the same as chocolate brownies?
- corrected to Sponge cakes.
Reviewer 2 Report
The introduction is generally well written, and the topic of the paper will be of interest to the readers of this journal. I have concerns regarding the structure of the methods section (as well as some questions on the methods to obtain sensory data). I have included some suggestions for the authors below to improve the clarity of the methods and results section.
Methods:
Perhaps the authors could consider including a brief study design section at the start of the materials and methods section? This will be clearer to the reader on the order and design of the study. It will also be helpful here to include a figure on the order of the study (i.e., Acceptance test on x number of consumers; Optimised Descriptive Profiling on x number of panellists, texture?).
Lines 45-56: It will be clearer here to include some statistics. For example, how many % of added sugar products (in particularly baked goods) contributes to the dietary intake of sugar and fat in developed countries? What is the average energy % dietary intake of sugar and fat?
Lines 63-64: Perhaps include the references here too?
Line 116: Spelling typo for “Australia”
Lines 124-130: Could the authors please expand the method to determine iso-sweetness further? It is unclear to me how a ranking test can be used to determine the concentration of Reb A needed to replace the sweetness concentration of sucrose. If the authors meant ranking a set of Reb A solutions from low intensity to high intensity, how are the authors able to match the sweetness concentration as ranking only measures the relative sweetness intensity within the set of samples (not in comparison to other set of sweeteners)? Did your assessors evaluate the sweetness intensity of the samples on an intensity scale?
Lines 142-144: It will be clearer to direct the readers to section 2.4 here (on Sensory Acceptance Testing).
Lines 145-147: The 4 formulations prepared where fat was replaced was based on which sucrose replacement sample (SC25/0, SC50/0…)? Has the formulation been adjusted based on the acceptance testing? It will be clearer to state them here too for the readers.
Lines 150-151: Did the researchers weigh the eggs? Eggs do vary in sizes and weight and may account for variation within products. It will be clearer to state them here too.
Line 169: Was “Chocolate brownie samples” a typo? Or was “Chocolate brownie samples” used as a training/dummy sample for rating liking?
Line 173: How was overall acceptability of samples determined here? Was this based on the highest average mean for the samples? Perhaps direct the reader to the statistical analysis section here too.
Section 2.6: I am not sure why the methods used to obtain photographs are included in the methods section. Are these important for the experimental design? Perhaps include a summary of these information as a figure description (Figure 3).
Line 291: Perhaps include a reference for this statement?
Results and discussion:
3.1 The authors need to describe clearly (As mentioned above) on how ranking data was obtained? Are these sweetness intensities mean scores (table 3)?
Figures 1 &2: The figure titles are incomplete “()”
Line 326: “The SC75/0 sample which is situated in the outer circle of the upper left quadrant, was very anti correlated with all liking parameters and OA.” Perhaps rephrase to “negatively correlated” instead of “anti-correlated”? What do the authors imply by “very anti correlated”? It will be clearer here to include examples or to explain the data further. Please edit throughout the results/discussion section.
Line 339: Perhaps just “significantly negatively associated” – removing “very”? Please edit throughout the results/discussion section.
Line 350: The 25% SR samples were only assumed to be similar in terms of the attributes measured. The authors did not conduct difference testing, so it cannot be assumed that the samples are similar (or the consumers are not able to distinguish between them). Please edit accordingly.
Line 408: I suggest the authors to consider including the actual p values here when describing the data. It will be clearer for the reader.
Line 544: “&” should be spelled out fully. Please edit this throughout (including abstract).
Lines 546-547: Please include a reference here. Are there references to support increased satiation/satiety levels after consumption of sweetened baked products with added inulin?
Author Response
Dear Reviewer 2,
thank you for your careful consideration of our manuscript and suggestions for its improvement. We tried to do our best to reply to your questions.
Perhaps the authors could consider including a brief study design section at the start of the materials and methods section? This will be clearer to the reader on the order and design of the study. It will also be helpful here to include a figure on the order of the study (i.e., Acceptance test on x number of consumers; Optimised Descriptive Profiling on x number of panellists, texture?).
- In Materials and Methods a new scheme was added and a phrase: “Sponge cakes research was followed by the represented methodology design scheme.”
(Scheme was added.)
Lines 45-56: It will be clearer here to include some statistics. For example, how many % of added sugar products (in particularly baked goods) contributes to the dietary intake of sugar and fat in developed countries? What is the average energy % dietary intake of sugar and fat?
In line 44 was added: “According to WHO (World Health organisation) guidelines (2011) adults and children should consume only 10% of calories from sugars in daily diet which is about 10-14 teaspoons of sugar. I Ireland this number achieved 14.6% which is 31.5% higher than the recommended dose by WHO [2].”
Lines 63-64: Perhaps include the references here too?
- A ref to a Review was added:
[11]. Struck S., Jaros D.,1 S. Brennan C. & Rohm H. (2014). Sugar replacement in sweetened bakery goods. Int. J. of Food Sci. and Tech., 49, 1963–1976.
Line 116: Spelling typo for “Australia”
- Corrected.
Lines 124-130: Could the authors please expand the method to determine iso-sweetness further? It is unclear to me how a ranking test can be used to determine the concentration of Reb A needed to replace the sweetness concentration of sucrose. If the authors meant ranking a set of Reb A solutions from low intensity to high intensity, how are the authors able to match the sweetness concentration as ranking only measures the relative sweetness intensity within the set of samples (not in comparison to other set of sweeteners)? Did your assessors evaluate the sweetness intensity of the samples on an intensity scale?
- ODP test to range in the iso-sweetness to adjust the intensity of experimental varients to the control worked very well. An intensity test is the method that we used. Apologise for the confusion to the reviewer say that that part of the manuscript was incorrect and that ODP was used. It was a cut and paste error from a previous draft on which the student worked simultaneously. Big error! Thank you very much for your attention you noticed it!
- From the Introduction RDA test was removed, marked in red in Introduction. In line 150 in Methods also corrected to ODP.
Lines 142-144: It will be clearer to direct the readers to section 2.4 here (on Sensory Acceptance Testing).
- We decided to leave it as it is because we highlighting that we selected the best sample and then put it further for fat replacements.
Lines 145-147: The 4 formulations prepared where fat was replaced was based on which sucrose replacement sample (SC25/0, SC50/0…)? Has the formulation been adjusted based on the acceptance testing? It will be clearer to state them here too for the readers.
- Yes, it was adjusted based on acceptance testing.
Lines 150-151: Did the researchers weigh the eggs? Eggs do vary in sizes and weight and may account for variation within products. It will be clearer to state them here too.
- No, eggs haven’t been weighed. We used similar eggs in size, an average type. Added this in the text.
Line 169: Was “Chocolate brownie samples” a typo? Or was “Chocolate brownie samples” used as a training/dummy sample for rating liking?
- Typo, corrected to sponge cakes samples.
Line 173. How was overall acceptability of samples determined here? Was this based on the highest average mean for the samples? Perhaps direct the reader to the statistical analysis section here too.
In new line 212 and in SAT and in line 229 in ODP was added: Statistician analysis method in shown in the section 2.7.
Section 2.6: I am not sure why the methods used to obtain photographs are included in the methods section. Are these important for the experimental design? Perhaps include a summary of these information as a figure description (Figure 3).
- I highlighted this information for replication of this method for other readers. It is essential to use the same focus distance for photography of the objects in order to get similar results. So I’d like to leave it as it is as an additional information.
Line 291: Perhaps include a reference for this statement?
Ref. was added after: Statistical significance for the relationships analysed by PLS were defined as P<0.05-0.01 (significant), P<0.01-0.001 (highly significant) and P<0.001 (extremely significant).
Results and discussion:
3.1 The authors need to describe clearly (As mentioned above) on how ranking data was obtained? Are these sweetness intensities mean scores (table 3)?
Yes table 3 presents mean scores as measures of intensity. Please see explanation above.
Figures 1 &2: The figure titles are incomplete “()”
- Corrected.
Line 326: “The SC75/0 sample which is situated in the outer circle of the upper left quadrant, was very anti correlated with all liking parameters and OA.” Perhaps rephrase to “negatively correlated” instead of “anti-correlated”? What do the authors imply by “very anti correlated”? It will be clearer here to include examples or to explain the data further. Please edit throughout the results/discussion section.
- Corrected to “negatively” correlated throughout all text.
Line 339: Perhaps just “significantly negatively associated” – removing “very”? Please edit throughout the results/discussion section.
- Deleted “very” throughout the text.
Line 350: The 25% SR samples were only assumed to be similar in terms of the attributes measured. The authors did not conduct difference testing, so it cannot be assumed that the samples are similar (or the consumers are not able to distinguish between them). Please edit accordingly.
They were similar based on non-significant differences based on their scores.
Line 408: I suggest the authors to consider including the actual p values here when describing the data. It will be clearer for the reader.
- Changed to: “significantly positively (p<0.05) associated”
Line 544: “&” should be spelled out fully. Please edit this throughout (including abstract).
- Corrected throughout the text.
Lines 546-547: Please include a reference here. Are there references to support increased satiation/satiety levels after consumption of sweetened baked products with added inulin?
- The phrase Therefore, overconsumption of these new developed products would be less likely due to increased satiety as a result of increased fibre content, which would contribute to a better caloric balance was deleted.